# Melioidosis–a disease of socioeconomic disadvantage

**Josh Hanson**[1,2]*, **Simon Smith**[2], **James Stewart**[2], **Peter Horne**[3], **Nicole Ramsamy**[4]

**1** Kirby Institute, University of New South Wales, Sydney, Australia, **2** Department of Medicine, Cairns Hospital, Cairns, Queensland, Australia, **3** Tropical Public Health Service, Cairns, Queensland, Australia, **4** Torres and Cape Hospital and Health Service, Cairns, Queensland, Australia

* jhanson@kirby.unsw.edu.au

**Data Availability Statement:** Data cannot be shared publicly because of the Queensland Public Health Act 2005. Data are available from the Far North Queensland Human Research Ethics Committee (contact via email

## Abstract

### Background

There is growing recognition of the contribution of the social determinants of health to the burden of many infectious diseases. However, the relationship between socioeconomic status and the incidence and outcome of melioidosis is incompletely defined.

### Methods

All residents of Far North Queensland, tropical Australia with culture-proven melioidosis between January 1998 and December 2020 were eligible for the study. Their demographics, comorbidities and socioeconomic status were correlated with their clinical course. Socioeconomic status was determined using the Socio-Economic Indexes for Areas (SEIFA) Index of Relative Socio-economic Disadvantage score, a measure of socioeconomic disadvantage developed by the Australian Bureau of Statistics. Socioeconomic disadvantage was defined as residence in a region with a SEIFA score in the lowest decile in Australia.

### Results

321 eligible individuals were diagnosed with melioidosis during the study period, 174 (54.2%) identified as Indigenous Australians; 223/321 (69.5%) were bacteraemic, 85/321 (26.5%) required Intensive Care Unit (ICU) admission and 37/321 (11.5%) died. 156/321 (48.6%) were socioeconomically disadvantaged, compared with 56603/269002 (21.0%) of the local general population (p<0.001). Socioeconomically disadvantaged patients were younger, more likely to be female, Indigenous, diabetic or have renal disease. They were also more likely to die prior to hospital discharge (26/156 (16.7%) versus 11/165 (6.7%), p = 0.002) and to die at a younger age (median (IQR) age: 50 (38–68) versus 65 (59–81) years, p = 0.02). In multivariate analysis that included age, Indigenous status, the presence of bacteraemia, ICU admission and the year of hospitalisation, only socioeconomic disadvantage (odds ratio (OR) (95% confidence interval (CI)): 2.49 (1.16–5.35), p = 0.02) and ICU admission (OR (95% CI): 4.79 (2.33–9.86), p<0.001) were independently associated with death.

Cairns_Ethics@health.qld.gov.au) for researchers who meet the criteria for access to confidential data.

**Funding:** The authors received no specific funding for this work.

**Competing interests:** The authors have declared that no competing interests exist.

## Conclusion

Melioidosis is disease of socioeconomic disadvantage. A more holistic approach to the delivery of healthcare which addresses the social determinants of health is necessary to reduce the burden of this life-threatening disease.

## Author summary

The social determinants of health—the circumstances in which people grow, live, work, and age, and the systems put in place to deal with illness—have a profound effect on how, when, and even if patients access healthcare, and yet they are generally under-appreciated by practicing clinicians whose training emphasises the biomedical model of healthcare. In this region of tropical Australia patients diagnosed with melioidosis were more likely to live in a socioeconomically disadvantaged region. Socioeconomically disadvantaged individuals with melioidosis were also more likely to die from their infection and to die at a younger age. It was notable that socioeconomic disadvantage had a greater independent association with in-hospital death than age, Indigenous status, the presence of bacteraemia or any of the comorbidities that classically predispose individuals to melioidosis. A more holistic approach to the delivery of healthcare—which addresses the social determinants of health—is necessary if we are to reduce the burden of melioidosis and the many other health conditions that disproportionately affect the most disadvantaged members of our society.

## Introduction

Melioidosis is caused by the environmental Gram-negative bacterium *Burkholderia pseudomallei* [1]. The organism lives in soil and surface water and is endemic to tropical Australia and Southeast Asia, but it has a wide global distribution [2]. Melioidosis is estimated to kill 89,000 people annually [2]. While some have suggested that the organism has the potential for use in bioterrorism [3,4], the risk of disease in the general population is actually relatively low, with only 1 in 4600 antibody producing exposures resulting in disease [5,6]. However, the disease is seen far more frequently in individuals with specific comorbidities, especially those living with diabetes mellitus, chronic lung disease or chronic kidney disease; people consuming alcohol in a hazardous manner are also at increased risk [5,7].

All these comorbidities are more common in the Indigenous Aboriginal and Torres Strait Islander peoples of Australia who experience a disproportionate burden of the disease [8]. The incidence of melioidosis is even higher among Indigenous Australians living in remote, socioeconomically disadvantaged locations, where environmental exposure to *B. pseudomallei* may be greater and where access to comprehensive health care is frequently limited [9,10]. In the tropical region of Far North Queensland (FNQ), not only is the incidence of melioidosis higher in Indigenous Australians, but so too is the mortality rate: between 1998 and 2016 the case-fatality rate was 19% compared with 6% among non-Indigenous patients [11]. Determining the explanation for this higher case-fatality rate is not straightforward. In cases of melioidosis, the interplay between host, pathogen and environment is complicated and, even in Australia's well-resourced health system, there are challenges in providing optimal access to care in remote locations [12,13].

However, there is a growing recognition of the contribution of socioeconomic disadvantage to disease incidence and health outcomes [14]. The social determinants of health—the circumstances in which people grow, live, work, and age, and the systems put in place to deal with illness—have a profound effect on how, when, and even if patients access healthcare, and yet they are generally under-appreciated by practicing clinicians whose training emphasises the biomedical model of healthcare [15,16].

This study was performed to determine the contribution of socioeconomic disadvantage to the incidence of melioidosis in FNQ and its influence on outcome. It was hoped that this would provide data that might inform public health strategies to reduce the local burden of disease.

## Methods

### Ethics statement

The Far North Queensland Human Research Ethics Committee provided ethical approval for the study (HREC/15/QCH/46±977) and waived the requirement for informed consent as the data were presented in an aggregated manner.

Cairns Hospital is a 531-bed, tertiary-referral public hospital located in the tropical far north of the state of Queensland, Australia. It served a population—in 2020—of approximately 280,000 people who live in an area of 380,000 $km^2$; 17% of the 2020 population identified as Indigenous Aboriginal or Torres Strait Islanders [17]. Patients were eligible for inclusion in the study if they had a positive culture for *B. pseudomallei* in the Cairns Hospital laboratory, the sole microbiology provider for FNQ health services, between January 1, 1998 (the first complete year after a local electronic laboratory database was established) and December 31, 2020.

Before January 2017, data were collected retrospectively, after this time data were collected prospectively. The patients' medical records were reviewed to determine their demographics, their clinical presentation, their co-morbidities, and their disease course as previously described [11]. The patients' Indigenous status was recorded; when individuals register with the public health system, they are routinely asked whether they identify as an Aboriginal Australian, a Torres Strait Islander Australian, both or neither. Hazardous alcohol use was said to be present if it had been documented in the medical record in the 12 months prior to presentation. Chronic lung disease was said to be present if a patient was receiving any ongoing treatment for a chronic lung condition. Chronic kidney disease was said to be present if there had been a serum creatinine >150 μmol/L documented before the presentation. Immunosuppression was said to be present if the patient was using immunosuppressive agents, including corticosteroids, chemotherapy, or immunomodulatory therapies. The presence of an active malignancy was also recorded. If a risk factor was not documented, it was presumed to be absent. Admission to the Intensive Care Unit (ICU) and death attributable to melioidosis prior to hospital discharge were also documented.

Australian Bureau of Statistics population data collected during the 2016 census were used to calculate disease incidence and prevalence [18,19]. If an individual lived in the region's administrative hub—the city of Cairns—or its surrounds, they were said to have an urban address, otherwise they were deemed to live in a rural or remote area. Socioeconomic status was quantified using the Socio-Economic Indexes for Areas (SEIFA) Index of Relative Socioeconomic Disadvantage score, a measure of socioeconomic disadvantage developed by the Australian Bureau of Statistics [20]. The score summarises a range of information about the economic and social conditions of people and households within an area and is calculated using the most recent census data. The most disadvantaged patients were defined as those

residing in an area with a SEIFA score in the lowest decile in Australia. The residential address of the patients with melioidosis was used to determine their SEIFA Index of Relative Socio-economic Disadvantage score at the Statistical Area Level 1 (SA1) level, calculated by Australian Bureau of Statistics using 2016 census data [21]. These scores were compared with the 2016 SA1 level Index of Relative Socio-economic Disadvantage scores of the entire FNQ region.

## Statistical analysis

Data were de-identified, entered into an electronic database (Microsoft Excel) and analysed using statistical software (Stata version 14.2). Groups were analysed using logistic regression, the Kruskal-Wallis, the chi-squared and Fisher's exact test, where appropriate. Trends over time were determined using an extension of the Wilcoxon rank-sum test. To determine the independent predictors of death, variables with a $p < 0.20$ in univariate analysis were included in a multivariate model. Multivariate analysis was then performed using backwards stepwise logistic regression, with variables only retained in the model if they had a $p < 0.05$.

# Results

There were 341 individual patients with a positive culture for *B. pseudomallei* during the study period; 321 (94.1%) were FNQ residents. The demographic characteristics, the recorded comorbidities, and the clinical course of these 321 patients are presented in Table 1.

## Trends in incidence and case-fatality rate

The incidence of melioidosis in FNQ increased from 4.1/100,000/year in 1998 to 15.0/100,000/year in 2020 (p for trend = 0.007). The case-fatality rate declined from 17/89 (19.1%) in the first half of the study period to 20/232 (8.6%) in the second half (p for trend = 0.008). This decline in case-fatality rate occurred despite there being no change in the proportion of

**Table 1. Selected characteristics of the cohort.**

| Variable | n = 321 |
|---|---|
| Age | 52 (42–64) |
| Children (age <18 years) | 12 (3.7%) |
| Male gender | 227 (70.7%) |
| Indigenous Australian | 174 (54.2%) |
| Rural/remote residence | 178 (55.5%) |
| Diabetes mellitus | 169 (52.7%) |
| Hazardous alcohol use | 117 (36.5%) |
| Chronic kidney disease | 49 (15.3%) |
| Chronic lung disease | 53 (16.5%) |
| Immunosuppression | 44 (13.7%) |
| Active malignancy | 26 (8.1%) |
| No documented risk factors | 46 (14.3%) |
| Current tobacco smoker | 156 (48.5%) |
| Bacteraemic | 223 (69.5%) |
| Intensive Care Unit admission | 85 (26.5%) |
| Death attributable to melioidosis | 37 (11.5%) |
| Age at death in the 37 individuals that died (years) | 54 (44–73) |

Values are median (interquartile range) or number (%)

**Table 2. Selected characteristics of the cohort stratified by rural/remote residence.**

|  | Urban residence n = 143 | Rural or remote residence n = 178 | p |
|---|---|---|---|
| Age > 50 years | 98 (68.5%) | 83 (46.6%) | <0.0001 |
| Children (age <18 years) | 4 (2.8%) | 8 (4.5%) | 0.56 |
| Male gender | 97 (67.8%) | 130 (73.0%) | 0.31 |
| Indigenous Australian | 49 (34.3%) | 125 (70.2%) | <0.0001 |
| Diabetes mellitus | 66 (46.2%) | 103 (57.9%) | 0.04 |
| Hazardous alcohol use | 44 (30.8%) | 73 (41.0%) | 0.06 |
| Chronic lung disease | 34 (23.8%) | 19 (10.7%) | 0.002 |
| Chronic kidney disease | 26 (18.2%) | 23 (12.9%) | 0.19 |
| Immunosuppression | 29 (20.3%) | 15 (8.4%) | 0.002 |
| Cancer diagnosis | 18 (12.6%) | 8 (4.5%) | 0.01 |
| Bacteraemic | 105 (73.4%) | 118 (66.3%) | 0.17 |
| Intensive Care Unit admission | 44 (30.8%) | 41 (23.0%) | 0.12 |
| Death attributable to melioidosis | 15 (10.5%) | 22 (12.4%) | 0.60 |
| Age at death in those that died (years) | 59 (47–76) | 51 (40–66) | 0.26 |

Values are median (interquartile range) or number (%)

bacteraemic cases (61/89 (68.5%) in the first half of the study period versus 162/232 (69.8%) in the second half, p = 0.53).

## Influence of residential address on disease incidence

Residents of rural and remote areas were over-represented: 178/321 (55.5%) in the cohort lived in a rural or remote area, compared with 112248/269002 (41.7%) of the FNQ population in the 2016 national census (p<0.0001). The residents of rural and remote areas with melioidosis in this cohort were younger and more likely to be Indigenous or diabetic. Urban residents were more likely to have chronic lung disease, a cancer diagnosis or to be immunosuppressed. However, there was no difference in case-fatality rate—or the age of death—between urban residents and those living in a rural/remote location (Table 2).

## Comparison of Indigenous and non-Indigenous Australians

Indigenous Australians were over-represented: 174/321 (54.2%) of the cohort were Indigenous Australians compared with 38785/265070 (14.6%) of the FNQ population in the national 2016 census (p<0.0001). Indigenous Australians were more likely to be female, younger or to have diabetes or renal disease than non-Indigenous Australians, although in this cohort they were less likely to have chronic lung disease, immunosuppression, or a cancer diagnosis. Indigenous patients were more likely to die than non-Indigenous patients (Table 3). They also died at a younger age (median (IQR) age: 47 (34–60) versus 72 (60–81) years, p = 0.0005).

## Influence of socioeconomic disadvantage

Patients in this cohort were more socioeconomically disadvantaged than the general FNQ population: 156/321 (48.6%) of the melioidosis cases had a SEIFA score in the lowest decile, compared with compared with 56603/269002 (21.0%) of the general FNQ population (p<0.001) (Fig 1). Indigenous Australians were more likely than non-Indigenous Australians to live in a region with the lowest decile SEIFA score (odds ratio (95% confidence interval): 13.8

**Table 3. Selected characteristics of the cohort stratified by Indigenous status.**

| | Non-Indigenous Australian n = 147 | Indigenous Australian n = 174 | p |
|---|---|---|---|
| Age > 50 years | 109 (74.2%) | 72 (41.4%) | <0.0001 |
| Child (age <18 years) | 4 (2.7%) | 8 (4.6%) | 0.56 |
| Male gender | 118 (80.3%) | 109 (62.6%) | <0.0001 |
| Rural/remote residence | 53 (36.1%) | 125 (71.8%) | <0.0001 |
| Diabetes mellitus | 53 (36%) | 116 (67%) | <0.0001 |
| Hazardous alcohol use | 53 (36%) | 64 (37%) | 0.89 |
| Chronic lung disease | 39 (26.5%) | 14 (8.1%) | <0.0001 |
| Chronic kidney disease | 13 (8.8%) | 36 (20.7%) | 0.003 |
| Immunosuppression | 27 (18.4%) | 17 (9.8%) | 0.03 |
| Cancer diagnosis | 18 (12.2) | 8 (4.6%) | 0.01 |
| Bacteraemic | 97 (66.0%) | 126 (72.4%) | 0.21 |
| Intensive Care Unit admission | 34 (23.1%) | 51 (29.3%) | 0.21 |
| Death attributable to melioidosis | 11 (7.5%) | 26 (14.9%) | 0.04 |
| Age at death in those that died (years) | 72 (60–81) | 47 (34–60) | 0.0005 |

Values are median (interquartile range) or number (%)

(8.0–23.7), p<0.001). Of the 146 non-Indigenous Australians in the cohort 26 (17.7%) had a SEIFA score in the lowest decile (Fig 2).

Patients in this cohort with a SEIFA score in the lowest decile were younger, more likely to be female, Indigenous, diabetic or to have renal disease (Table 4). They were also more likely to die and to die at a younger age (median (IQR) age: 50 (38–68) compared with 65 (59–81) years, p = 0.02).

## Predictors of outcome

The association between selected characteristics and death are presented in Table 5; Indigenous status, socioeconomic disadvantage and ICU admission were all significantly associated with outcome in univariate analysis. In a multivariate model that included all variables with a

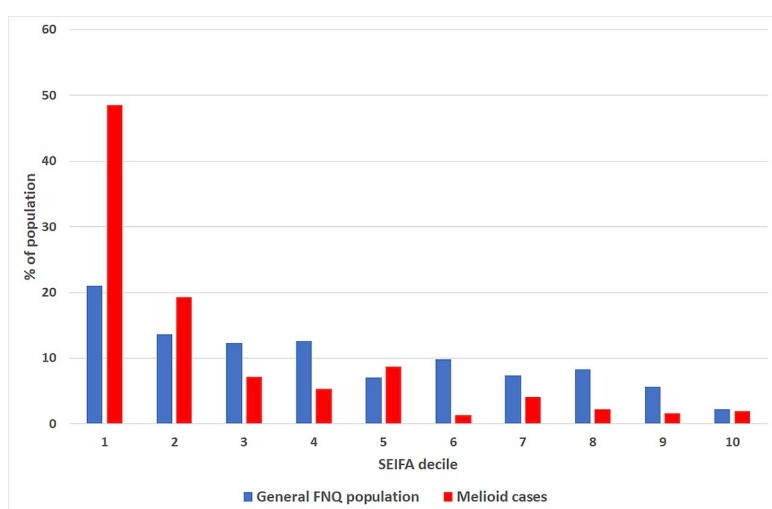

**Fig 1. Socioeconomic disadvantage of people with melioidosis (determined using the SEIFA Index of Relative Socio-economic Disadvantage score) compared with the general FNQ population.**

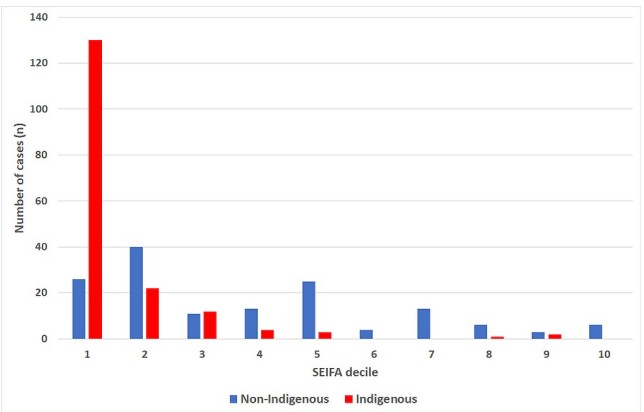

**Fig 2. Socioeconomic disadvantage of people with melioidosis (determined using the SEIFA Index of Relative Socio-economic Disadvantage score), stratified by Indigenous status.**

p value of <0.20 in univariate analysis and the year of hospitalisation (to account for the declining case-fatality rate over the study period), only socioeconomic disadvantage and ICU admission were associated with death (Table 6).

The case-fatality rate in each of the SEIFA deciles is presented in Fig 3. The proportion of the cohort's cases and deaths stratified by Indigenous status and SEIFA decile is presented in Fig 4.

## Discussion

Socioeconomic disadvantage significantly increases the risk of melioidosis in this region of tropical Australia. It also has a greater association with outcome than age or any of the comorbidities that are classically associated with the disease. While work towards a vaccine for *B. pseudomallei* continues, a greater focus on addressing the social determinants of health in

**Table 4. Association between selected characteristics of the cohort and socioeconomic disadvantage (determined by SEIFA Index of Relative Socio-economic Disadvantage score).**

|  | Lowest decile SEIFA score n = 156 | SEIFA score in deciles 2–10 n = 165 | p |
|---|---|---|---|
| Age > 50 years | 69 (44.2%) | 112 (67.9%) | <0.0001 |
| Child (age <18 years) | 8 (5.1%) | 4 (2.4%) | 0.25 |
| Male gender | 97 (62.2%) | 130 (78.8%) | 0.001 |
| Rural/remote residence | 111 (71.2%) | 67 (40.6%) | <0.0001 |
| Indigenous Australian | 130 (83.3%) | 44 (26.7%) | <0.0001 |
| Diabetes mellitus | 100 (64.1%) | 69 (41.8%) | <0.0001 |
| Hazardous alcohol use | 58 (37.2%) | 59 (35.8%) | 0.79 |
| Chronic lung disease | 18 (11.5%) | 35 (21.2%) | 0.02 |
| Chronic kidney disease | 31 (19.9%) | 18 (10.9%) | 0.03 |
| Immunosuppression | 15 (9.6%) | 29 (17.6%) | 0.04 |
| Cancer diagnosis | 6 (3.9%) | 20 (12.1%) | 0.007 |
| Bacteraemic | 111 (71.2%) | 112 (67.9%) | 0.52 |
| Intensive Care Unit admission | 49 (31.4%) | 36 (21.8%) | 0.05 |
| Death from melioidosis prior to hospital discharge | 26 (16.7%) | 11 (6.7%) | 0.002 |
| Age at death in those that died (years) | 50 (38–68) | 65 (59–81) | 0.02 |

Values are median (interquartile range) or number (%)

**Table 5. Association between selected characteristics of cases of melioidosis and death.**

|  | Died n = 37 | Survived n = 284 | p |
|---|---|---|---|
| Age > 50 years | 22 (59.5%) | 159 (56.0%) | 0.69 |
| Child (age <18 years) | 3 (8.1%) | 9 (3.2%) | 0.15 |
| Male gender | 25 (67.6%) | 202 (71.1%) | 0.66 |
| Rural/remote residence | 22 (59.5%) | 156 (54.9%) | 0.60 |
| Indigenous Australian | 26 (70.3%) | 148 (52%) | 0.04 |
| Lowest decile SEIFA score | 26 (70.3%) | 130 (45.8%) | 0.005 |
| Diabetes | 16 (43.2%) | 153 (53.9%) | 0.22 |
| Hazardous alcohol use | 13 (35.1%) | 104 (36.6%) | 0.86 |
| Chronic lung disease | 8 (21.6%) | 45 (15.9%) | 0.37 |
| Chronic kidney disease | 8 (21.6%) | 41 (14.4%) | 0.25 |
| Immunosuppression | 5 (13.5%) | 39 (13.7%) | 0.97 |
| Cancer diagnosis | 5 (13.5%) | 21 (7.4%) | 0.20 |
| Bacteraemic | 30 (81.1%) | 193 (68.0%) | 0.10 |
| Intensive Care Unit admission | 22 (59.5%) | 63 (22.2%) | <0.001 |

Values are presented as absolute number (%)

Australia is likely to not only reduce the morbidity and mortality related to melioidosis but would also alleviate the impact of the many other health conditions that disproportionately affect the most disadvantaged members of this wealthy country [22,23].

More than half of the melioidosis cases in the cohort occurred in Indigenous Australians, despite their representing only 14.6% of the local population in the most recent national census. This finding is not novel, nor is the fact that two-thirds of the Indigenous patients in the series had diabetes, the most common risk factor for melioidosis in this cohort and many others [7,24,25]. However, previous Australian series have not quantified the relative contribution of socioeconomic disadvantage to disease incidence and outcome. It was notable, therefore, that almost half of all the patients in this cohort lived in the most disadvantaged districts in the country (those with a SEIFA score in the lowest decile) and that the only independent predictors of death in the series were socioeconomic status and a requirement for ICU care, a proxy for severe disease.

This is an important point as previous Australian studies—which had a similar mix of predisposing conditions to this cohort—have reported that the presence of these predisposing comorbidities—and older age—are the strongest predictors of outcome [11,24]. This current study's data lead to the greater truth that while almost all the patients in this cohort presented acutely, their clinical presentation and death from melioidosis was, in many cases, decades in the making. Despite Australia's universal health system, this finding is not unique to

**Table 6. Multivariate analysis to identify factors independently associated with death.**

|  | Univariate analysis | | Multivariate analysis | |
|---|---|---|---|---|
|  | Odds ratio (95%CI) | p | Odds ratio (95%CI) | p |
| Indigenous Australian | 2.17 (1.03–4.56) | 0.04 | 1.20 (0.47–3.09) | 0.70 |
| Lowest decile SEIFA score | 2.80 (1.33–5.88) | 0.007 | 2.49 (1.16–5.35) | 0.02 |
| Bacteraemia | 2.02 (0.86–4.77) | 0.11 | 1.30 (0.52–3.28) | 0.58 |
| Intensive Care Unit admission | 5.14 (2.52–10.50) | <0.001 | 4.79 (2.32–9.86) | <0.001 |
| Year of hospitalisation | 0.94 (0.90–0.99) | 0.01 | 0.96 (0.91–1.01) | 0.11 |

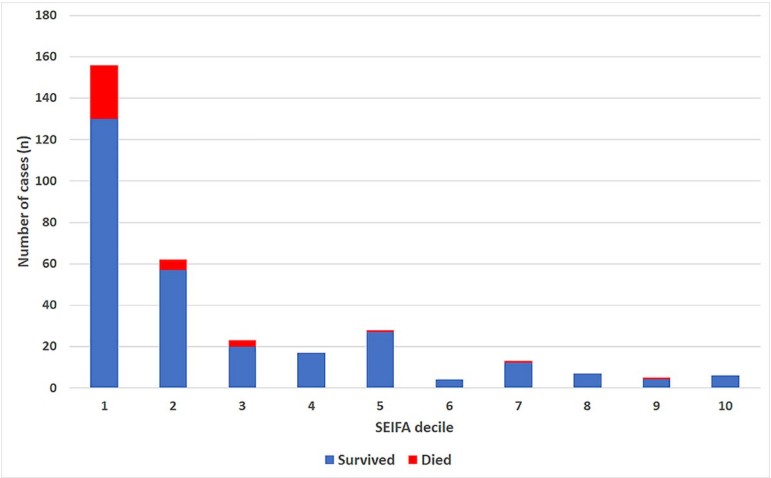

**Fig 3. Case-fatality rate in the cohort stratified by SEIFA decile.**

melioidosis; an association with socioeconomic disadvantage is seen in a variety of infectious diseases and non-communicable diseases seen in the region [13,26–32].

Although Aboriginal and Torres Strait Islander status did not have an independent association with outcome in this study, the cohort's Indigenous Australians were more than three times as likely to live in a region with a SEIFA score in the lowest decile. Indigenous Australians have poorer health outcomes than non-Indigenous Australians on almost every health metric and social determinants of health explain between one third and one half of this difference [9]. Successive Australian governments have attempted to address this complex issue, however, there has been very limited success [33]. It is now understood that greater engagement and shared decision making with Aboriginal and Torres Strait Islander communities is necessary if progress is to be made [34]. This is more likely to deliver holistic care that considers the social and economic context of the patient, that addresses social and emotional wellbeing, and which is provided in a culturally safe manner. The presence and integration of Aboriginal and Torres Strait Islander staff across the entire care system is also critical [35]. Although there are significant challenges in delivering this care to remote Indigenous communities, there are now several Australian regions where community-based and community-led healthcare programmes have translated into improved access to care and a reduced requirement for emergency care [36,37]. However, there is clearly still much to be done.

The case-fatality rate of melioidosis fell significantly during the study period and is now one of the lowest reported in the world. This is likely due to a variety of factors including earlier recognition of sepsis, evolution of the local aeromedical retrieval service and improvements in critical care [11,38]. However, the local incidence of melioidosis is rising and a more integrated approach to disease prevention and management is clearly necessary. Whilst a hub and spoke model has worked well in delivering specialist medical assessment and treatment in FNQ [29,39], providing interventions that prevent melioidosis in the geographically dispersed population is more complicated. Public health strategies that focus on human behaviour and aim to prevent exposure to *B. pseudomallei* are challenging to implement [40,41]. Chemoprophylaxis would not be cost-effective locally and is associated with significant side effects [42]. More comprehensive primary healthcare, with a focus on addressing the comorbidities that predispose individuals to melioidosis, would not only be expected to reduce the incidence of melioidosis [6], but would have the added benefit of reducing the burden of diseases that are

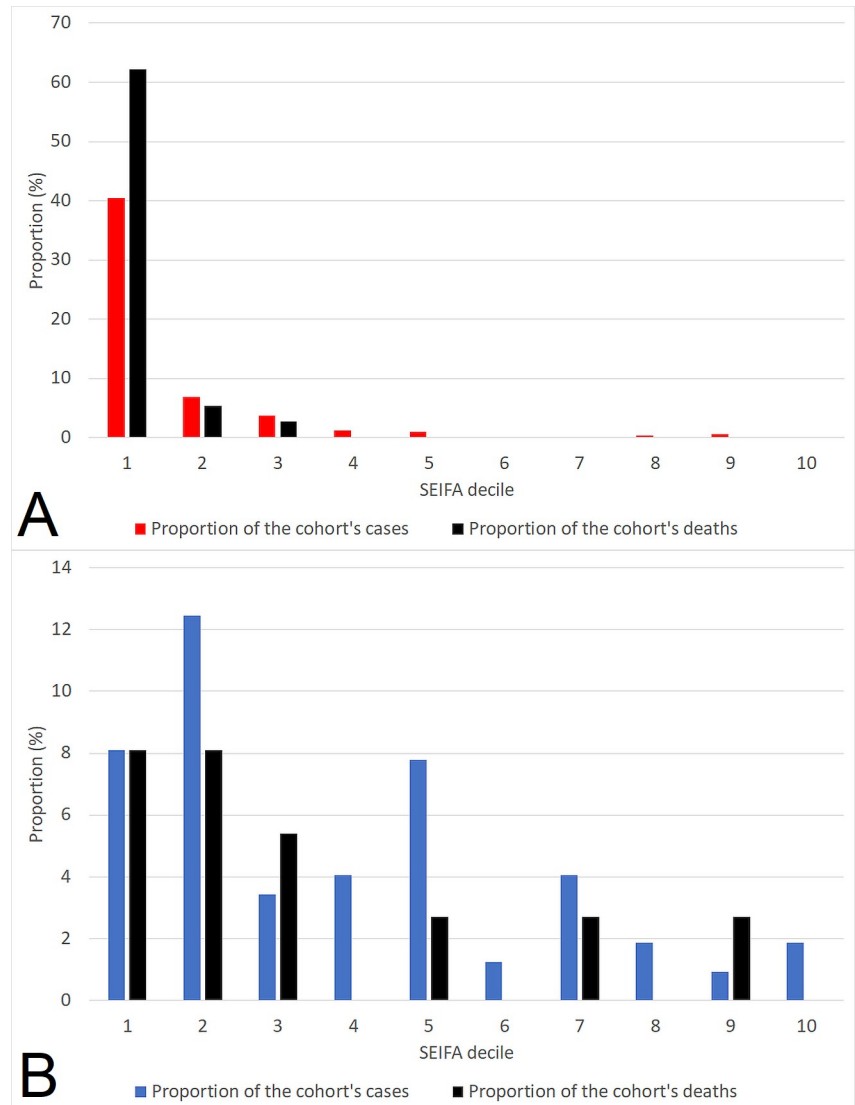

**Fig 4. Proportion of the cohort's cases and deaths stratified by Indigenous status (Indigenous patients presented in panel A, non-Indigenous patients presented in panel B).**

far more common causes of death locally [27,43,44]. Strengthening primary healthcare in the region would also be expected to improve the long-term prognosis of patients who survive melioidosis. a failure to deliver longitudinal, multidisciplinary care post-discharge has translated into high rates of premature and potentially reversible deaths in the region, with death occurring a median of 3.8 years after discharge at a mean age of 59 years [45].

The study has several limitations. Most of the data were collected retrospectively, accordingly some of these data—especially those related to comorbidities—are likely to be incomplete. This would have the effect of overestimating the proportion of cases in which no risk factor was identified. The study's retrospective design also precluded accurate documentation of symptom duration before presentation, which may have provided insights into patients' access to care. Although the SEIFA score is determined by the Australian Bureau of Statistics using census data, it is calculated for entire regions, so the scores do not necessarily reflect the social disadvantage of individual patients. SEIFA scores in remote locations are generally

lower than those in urban locations and so the association with socioeconomic disadvantage that is described might be hypothesised to simply result from greater exposure to *B. pseudomallei* and less access to comprehensive healthcare in rural and remote locations. However, while the incidence of melioidosis was higher in rural and remote locations, it was notable that the case-fatality rate was not significantly greater in residents of these regions. Indeed, it was striking that in multivariate analysis that—apart from a requirement for ICU care—socioeconomic disadvantage was the only factor associated with outcome. Different methods used by the Australian Bureau of Statistics to define the FNQ population led to minor variations in the size of the local population in the analyses that compared characteristics of the cohort to that of the general population, although this would not be expected to influence the study's findings [18,19]. Finally, while this study's findings are likely to be similar in other Australian jurisdictions [24], they may not necessarily be replicated in other countries, although it is certainly the case that relatively poor agricultural workers living in other parts of the world appear to have a greater burden of disease [46–48].

This study provides data that might be used to inform a more holistic approach to reduce the local burden of melioidosis. In essence, these data show that melioidosis is a disease that is predominantly borne by the disadvantaged and is, in some respects, an indicator of disadvantage itself. The case-fatality rate of melioidosis in the region continues to decline—and is now one of the lowest reported in the world—but until the underlying socioeconomic inequality that increases the risk of the disease is addressed, its burden will be unnecessarily great.

## Author Contributions

**Conceptualization:** Josh Hanson.

**Data curation:** Josh Hanson, Simon Smith, James Stewart, Peter Horne.

**Formal analysis:** Josh Hanson, Peter Horne.

**Investigation:** Josh Hanson, Simon Smith, Peter Horne, Nicole Ramsamy.

**Methodology:** Josh Hanson, Peter Horne.

**Supervision:** Nicole Ramsamy.

**Validation:** Josh Hanson, Simon Smith, Peter Horne, Nicole Ramsamy.

**Visualization:** Josh Hanson, Nicole Ramsamy.

**Writing – original draft:** Josh Hanson.

**Writing – review & editing:** Josh Hanson, Simon Smith, James Stewart, Peter Horne, Nicole Ramsamy.

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
