## [Decision Letter · Decision Letter 0]

24 May 2021

Dear Dr Hanson,

Thank you very much for submitting your manuscript "Melioidosis – a disease of socioeconomic disadvantage" for consideration at PLOS Neglected Tropical Diseases. As with all papers reviewed by the journal, your manuscript was reviewed by members of the editorial board and by several independent reviewers. The reviewers appreciated the attention to an important topic. Based on the reviews, we are likely to accept this manuscript for publication, providing that you modify the manuscript according to the review recommendations. 

Thank you for your manuscript. In addition to the reviewers comments, I had a few comments/questions that I was hoping your team could address. Your manuscript is timely and important - these comments are made with the hope of strengthening it. 

1) Have the diagnostic methods/yield for melioidosis changed over the period of your study? Could the decline in mortality over time be due to detection of less sick cases? Are you sure that better treatment is the reason for improved outcomes? Could it be earlier presentation? 

2) How do the outcomes change by SEIFA score deciles within Indigenous Australians? And between Indigenous and non-Indigenous Australians at different deciles? This may help tease apart the interaction between Indigenous and non-Indigenous as well as socio-economic status. 

3) I would urge caution in assessing causality from this retrospective study; there are may be confounders that were not assessed. For example chronic lung disease, cancer, immunosuppression were all higher in your urban, non-Indigenous, deciles 2-10; those are all risk factors for worse outcome. Can you include duration of symptoms prior to presenation as a predictor of outcomes? 

4) Would you be able to provide more context to your results - how do risk factors and outcomes compare with studies of melioidosis in other parts of Australia? Would you be able to use population adjusted rates, especially for discussions of mortality ( I do notice that it was done for incidence). If your study design is unique , could you please highlight in what ways it differs from previous studies? 

5) 1) Please shorten your discussion - your data prove the larger theme of your paper quite adequately; a shorter discsussion would definitely strengthen your manuscript. The importance of the social determinants are well known, as are the inequities that Indigenous Australians suffer, and the challenges in reducing them. Would you be able to distil it down to what is relevant to your results? And (most respectfully), please try not to editorialize in your discussion and avoid making broad statements.

Sincerely,

Husain Poonawala

Associate Editor

Elsio Wunder Jr

Deputy Editor

Thank you for your manuscript. In addition to the reviewers comments, I had a few comments/questions that I was hoping your team could address. Your manuscript is timely and important - these comments are made with the hope of strengthening it. 

1) Have the diagnostic methods/yield for melioidosis changed over the period of your study? Could the decline in mortality over time be due to detection of less sick cases? Are you sure that better treatment is the reason for improved outcomes? Could it be earlier presentation? 

2) How do the outcomes change by SEIFA score deciles within Indigenous Australians? And between Indigenous and non-Indigenous Australians at different deciles? This may help tease apart the interaction between Indigenous and non-Indigenous as well as socio-economic status. 

3) I would urge caution in assessing causality from this retrospective study; there are may be confounders that were not assessed. For example chronic lung disease, cancer, immunosuppression were all higher in your urban, non-Indigenous, deciles 2-10; those are all risk factors for worse outcome. Can you include duration of symptoms prior to presenation as a predictor of outcomes? 

4) Would you be able to provide more context to your results - how do risk factors and outcomes compare with studies of melioidosis in other parts of Australia? Would you be able to use population adjusted rates, especially for discussions of mortality ( I do notice that it was done for incidence). If your study design is unique , could you please highlight in what ways it differs from previous studies? 

5) 1) Please shorten your discussion - your data prove the larger theme of your paper quite adequately; a shorter discsussion would definitely strengthen your manuscript. The importance of the social determinants are well known, as are the inequities that Indigenous Australians suffer, and the challenges in reducing them. Would you be able to distil it down to what is relevant to your results? And (most respectfully), please try not to editorialize in your discussion and avoid making broad statements.

Reviewer's Responses to Questions

**Key Review Criteria Required for Acceptance?**

**Methods**

-Are the objectives of the study clearly articulated with a clear testable hypothesis stated?

-Is the study design appropriate to address the stated objectives?

-Is the population clearly described and appropriate for the hypothesis being tested?

-Is the sample size sufficient to ensure adequate power to address the hypothesis being tested?

-Were correct statistical analysis used to support conclusions?

-Are there concerns about ethical or regulatory requirements being met?

Reviewer #1: Yes

Reviewer #2: Are the objectives of the study clearly articulated with a clear testable hypothesis stated?

Yes

-Is the study design appropriate to address the stated objectives?

Yes

-Is the population clearly described and appropriate for the hypothesis being tested?

Yes

-Is the sample size sufficient to ensure adequate power to address the hypothesis being tested?

Yes

-Were correct statistical analysis used to support conclusions?

Yes

-Are there concerns about ethical or regulatory requirements being met?

No

Reviewer #3: No major concerns. The study methods are well-described and appropriate.

**Results**

-Does the analysis presented match the analysis plan?

-Are the results clearly and completely presented?

-Are the figures (Tables, Images) of sufficient quality for clarity?

Reviewer #1: Yes

Reviewer #2: Does the analysis presented match the analysis plan?

Yes

-Are the results clearly and completely presented?

Yes

-Are the figures (Tables, Images) of sufficient quality for clarity?

Yes

Reviewer #3: No concerns. The results are clearly presented.

**Conclusions**

-Are the conclusions supported by the data presented?

-Are the limitations of analysis clearly described?

-Do the authors discuss how these data can be helpful to advance our understanding of the topic under study?

-Is public health relevance addressed?

Reviewer #1: Yes

Reviewer #2: Are the conclusions supported by the data presented?

Yes

-Are the limitations of analysis clearly described?

Yes

-Do the authors discuss how these data can be helpful to advance our understanding of the topic under study?

Yes

-Is public health relevance addressed?

Yes

Reviewer #3: Again, no concerns. The conclusions are supported by the data and the discussion is very appropriate.

**Editorial and Data Presentation Modifications?**

Reviewer #1: (No Response)

Reviewer #2: (No Response)

Reviewer #3: This is a well designed study with suitable analysis and presentation of results. 

I have only a few minor comments:

It would be interesting to see a brief presentation and discussion of any data relating to delays in access to care for subjects in the study e.g. distance (space and time) from health facilities, or time to diagnosis of melioidosis after presentation. In other countries, the time to access medical care has an influence on mortality rates.

Urban/rural and remote might be better defined in the Methods - e.g. would Innisfail, Palm Cove or Port Douglas count as rural areas?

Could the authors discuss the possible causes for the declining case-fatality rate over the course of the study, and how this might relate to socioeconomic disadvantage?

**Summary and General Comments**

Reviewer #1: This is a well written and interesting paper that demonstrates a clear association between indicators of population disadvantage and both incidence and mortality from melioidosis. My only questions are whether the authors would be able to include any individual-specific data about socio-economic status and disadvantage, and also whether it would be possible to include an analysis of duration of symptoms prior to admission to hospital as a possible explanation for the observed mortality difference?

Reviewer #2: There is little information about the socioeconomic impact on melioidosis. This paper revealed that socioeconomically disadvantaged patients were more likely to have melioidosis and associated with mortality. The study suggests a holistic approach to the delivery of healthcare which addresses the social determinants of health is important, to reduce the burden of melioidosis as well as other diseases.

The topic is important and relevance to neglected tropical infections like melioidosis. It will be of interest for policy maker and researches. The conclusion is justified based on the current data. The study raises the questions whether this finding would be true in other counties.

Reviewer #3: This is a good study, which will be of considerable relevance for public health management of melioidosis (and other infectious diseases).

PLOS authors have the option to publish the peer review history of their article (what does this mean?). If published, this will include your full peer review and any attached files.

Reviewer #1: No

Reviewer #2: No

Reviewer #3: No

Figure Files:

Data Requirements:

Reproducibility:

References

---

## [Editor Report · Decision Letter 1]

7 Jun 2021

Dear Dr Hanson,

We are pleased to inform you that your manuscript 'Melioidosis – a disease of socioeconomic disadvantage' has been provisionally accepted for publication in PLOS Neglected Tropical Diseases.

Best regards,

Husain Poonawala

Associate Editor

Elsio Wunder Jr

Deputy Editor

Thank you for responding to the comments from the editor and reviewers.

Re: comment 3 -  " I would urge caution in assessing causality from this retrospective study; there are may be confounders that were not assessed. For example chronic lung disease, cancer, immunosuppression were all higher in your urban, non-Indigenous, deciles 2-10; those are all risk factors for worse outcome" - I was trying to highlight that despite having risks for worse outcomes that are consistent with our biological understanding of the disease, this groups did not have a worse outcome. My concern that this paradox could be due to unmeasured confounding that may arise from the retrospective nature of your data.

Re: comment 4 -  The discussion is much improved and balanced. We do not expect you to shy away from discussing important social determinants of health; we only request that it be related to the data presented in the results of the manuscript.

We look forward to sharing your valuable work with readers of PLoS NTD.

---

## [Editor Report · Acceptance letter]

16 Jun 2021

Dear Dr Hanson,

We are delighted to inform you that your manuscript, "Melioidosis – a disease of socioeconomic disadvantage," has been formally accepted for publication in PLOS Neglected Tropical Diseases.

Best regards,

Shaden Kamhawi

co-Editor-in-Chief

Paul Brindley

co-Editor-in-Chief
